# Automatic identification of species with neural networks

Andrés Hernández-Serna[1,2] and Luz Fernanda Jiménez-Segura[1]

[1] Grupo de Ictiología, Instituto de Biología, Universidad de Antioquia, Medellín, Colombia
[2] Department of Biology, University of Puerto Rico-Río Piedras, San Juan, PR, USA

## ABSTRACT

A new automatic identification system using photographic images has been designed to recognize fish, plant, and butterfly species from Europe and South America. The automatic classification system integrates multiple image processing tools to extract the geometry, morphology, and texture of the images. Artificial neural networks (ANNs) were used as the pattern recognition method. We tested a data set that included 740 species and 11,198 individuals. Our results show that the system performed with high accuracy, reaching 91.65% of true positive fish identifications, 92.87% of plants and 93.25% of butterflies. Our results highlight how the neural networks are complementary to species identification.

## INTRODUCTION

The Global Taxonomy Initiative highlights the knowledge gaps in our taxonomic system due to the shortage of trained taxonomists and curators which reduces our ability to understand, use, and conserve biological diversity (*Convention on Biological Diversity, 2014*). High levels of global biodiversity and a limited number of taxonomists represents significant challenges to the future of biological study and conservation. The main problem is that almost all taxonomic information exists in languages and formats not easily understood or shared without a high level of specialized knowledge and vocabularies. Thus, taxonomic knowledge is localized within limited geographical areas and among a limited number of taxonomists. This lack of accessibility of taxonomic knowledge to the general public has been termed the "taxonomic crisis" (*Dayrat, 2005*).

Recently, taxonomists have been searching for more efficient methods to meet species identification requirements, such as developing digital image processing and pattern recognition techniques. Researchers currently have recognition techniques for insects, plants, spiders, and plankton (*Gaston & O'Neill, 2004*). This approach can be extended even further to field-based identification of organisms such as fish (*Strachan, Nesvadba & Allen, 1990*; *Storbeck & Daan, 2001*; *White, Svellingen & Strachan, 2006*; *Zion et al., 2007*; *Hu et al., 2012*), insects (*Mayo & Watson, 2007*; *O'Neill, 2007*; *Kang, Song & Lee, 2012*), zooplankton (*Grosjean et al., 2004*) and plants (*Novotny & Suk, 2013*). These methods are helpful in alleviating the "taxonomy crisis". In this research, we present a new methodology for the identification of different taxonomic groups to the species level for fish, plants, and butterflies.

Corresponding author
Andrés Hernández-Serna,
andres137@gmail.com

We designed a simple and effective algorithm (preprocess solution) and defined a range of new features that use pattern recognition with artificial neural network designs (ANN).

## MATERIALS AND METHODS

### Images

Image data were taken from two sources: natural history museum records, and online (Data S1). Each collection was analyzed according to the country of origin. Ichthyology collections from Colombia were compiled from the Instituto de Investigaciones Marinas y Costeras (INVEMAR), the Colección de Referencia Biología Marina Universidad del Valle (CRBMUV), and the Coleccion Ictiologica Universidad de Antioquia (CIUA). Ichthyology collections from Brazil were found in the Museu de Zoologia da USP (MZUSP), the Instituto Nacional de Pesquisas da Amazônia Manaus (INPA), and the Museu Nacional Rio de Janeiro (MNRJ). Image data from Spain came from the Museo Nacional de Ciencias Naturales Madrid (MNCN). We tested a data set that included a total of 740 species and 11,198 individuals of fish, plants, and butterflies. Fish specimen images were taken using a Canon EOS 6dD one-use camera with a 1,280 × 960 pixel resolution. A total of 697 fish species previously identified by experts, were photographed (see Fig. 1 for a subset of photographed species). Images of 32 plant species were downloaded from the Flavia database (2009) (http://flavia.sourceforge.net/) (see Fig. 2). Image data for 11 species of butterflies were downloaded from the MorphBank database (Erickson et al., 2007) (see Fig. 3).

### System development

Based on pattern recognition theory (De Sá, 2001) and basic computer-processing pathways used in typical automated species identification systems (Gaston & O'Neill, 2004), we designed a system for automatic individual identification at the species level (Fig. 4). In a novel way, our system shared preprocess and extraction components with both training and recognition processes. Features of training images are used to build a model of the classification progress pattern after feature extraction. These features and the trained model were then recorded in the database and incorporated in the analysis of subsequent photos. This process used two types of data to model image features and resulted in better species identification results. Features can be found in Supplemental Information.

### Image preprocessing

Image heterogeneity in terms of orientation, size, brightness, and illumination was common (Fig. 5.1). The image background was removed with Grabcut's algorithm (Rother, Kolmogorov & Blake, 2004) (Fig. 5.2) and converted to grayscale (Fig. 5.3). Different filters were applied to improve the image by removing image noise; the filters used were smooth and median (Figs. 5.4 and 5.5), and the image was then reduced to one of two possible levels, 0 or 1 (Fig. 5.6). Next, the processed image was brought to a contour (Fig. 5.7) and then a skeleton (Fig. 5.8). All of these processes were performed for each taxonomic group using the image processing in MATLAB R2009b.

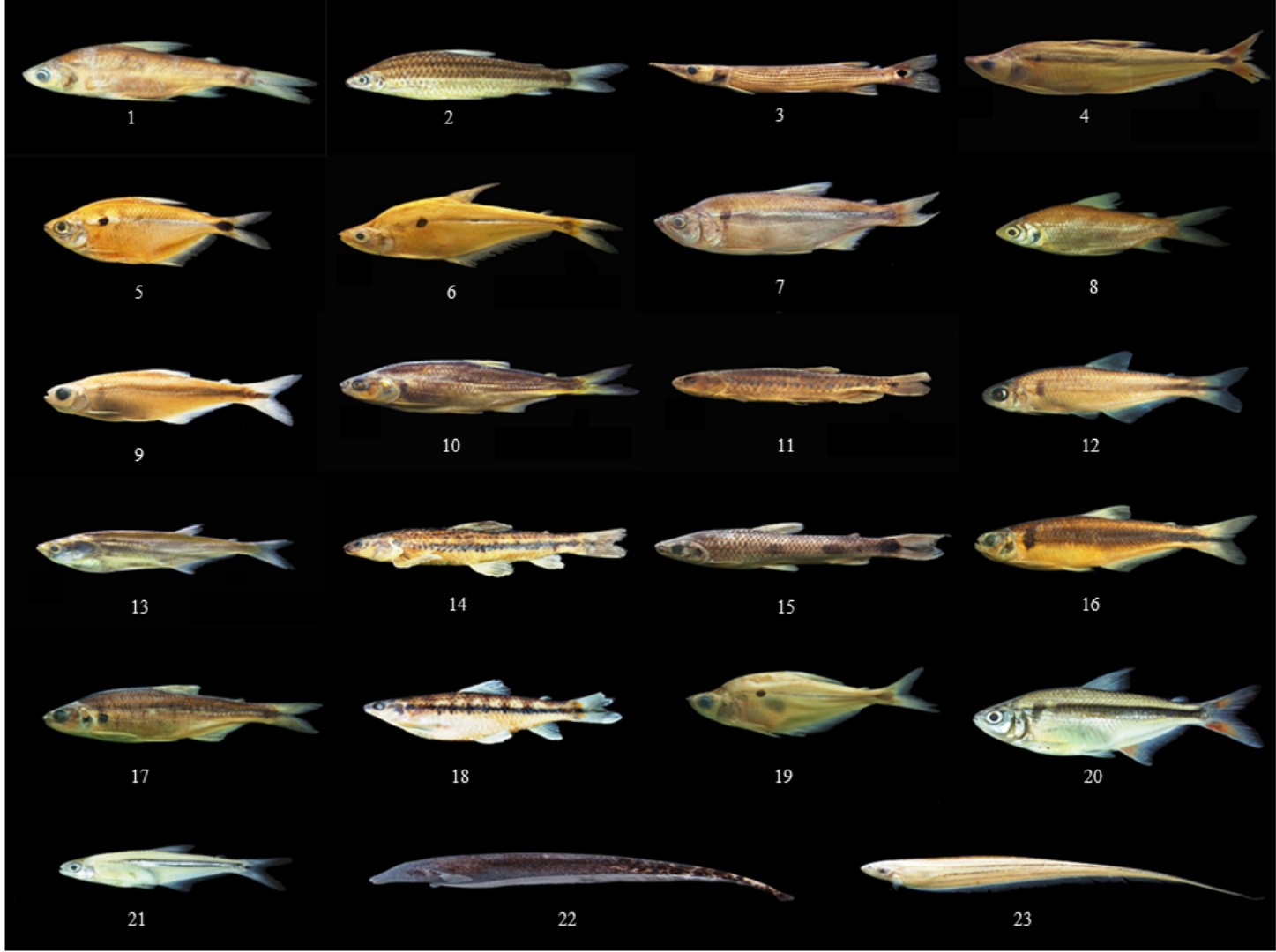

**Figure 1 Samples of some species.** (1) *Curimata mivartii* (2) *Leporinus striatus* (3) *Ctecolucius hujeta* (4) *Cinopotamus magdalenae* (5) *Astyanax magdalenae* (6) *Roeboides occidentalis* (7) *Genycharax tarpon* (8) *Cyphocharax magdalenae* (9) *Hemibrycon decurrens* (10) *Brycon medemi* (11) *Lebiasina multimaculata* (12) *Hemibrycon dentatus* (13) *Triporheus magdalenae* (14) *Characidium phoxocephalum* (15) *Leporinus muyscorum* (16) *Hemibrycon boquiae* (17) *Brycon hennir* (18) *Characidium caucanum* (19) *Roeboides dayi* (20) *Astyanax fasciatus* (21) *Argopleura magdalenensis* (22) *Apteronotus eschemeyeri* (23) *Eigenmannia virescens*.

## Feature extraction

A series of 15 geometrical, morphological, and texture features, that could be efficiently extracted with image processing and were unique to species, were used in our automatic identification system; these features can be efficiently extracted with image processing (Table 1).

## Geometrical

Geometric features contain information about form, position, size, and orientation of the region. The following six geometric features were commonly used in pattern recognition.

**Peer**J

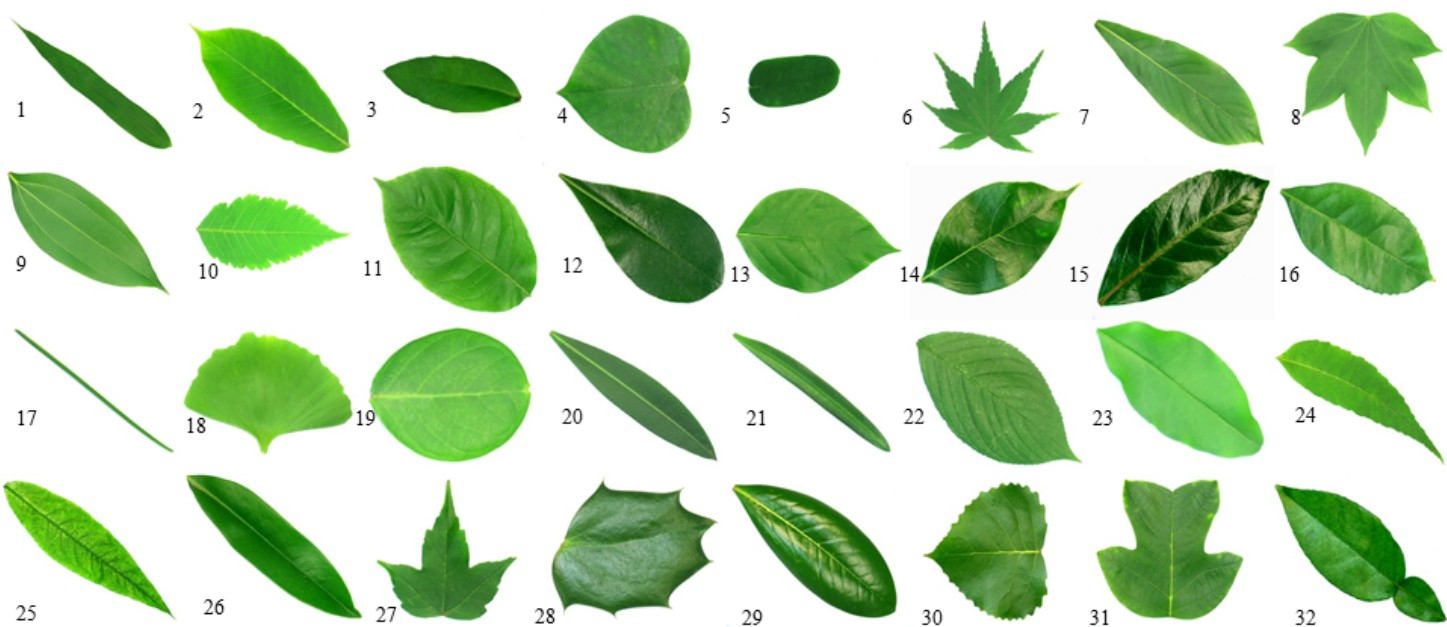

**Figure 2 Samples of plants.** (1) *Phyllostachys edulis* (2) *Aesculus chinensis* (3) *Berberis anhweiensis* (4) *Cercis chinensis* (5) *Indigofera tinctoria* (6) *Acer Dalmatum* (7) *Phoebe zhennan* (8) *Kalopanax septemlobus* (9) *Cinnamomum japonicum* (10) *Koelreuteria paniculata* (11) *Ilex macrocarpa* (12) *Pittosporum tobira* (13) *Chimonanthus praecox* (14) *Cinnamomum camphora* (15) *Viburnum awabuki* (16) *Osmanthus fragrans* (17) *Cedrus deodara* (18) *Ginkgo biloba* (19) *Lagerstroemia indica* (20) *Nerium oleander* (21) *Podocarpus macrophyllus* (22) *Prunus yedoensis* (23) *Ligustrum lucidum* (24) *Tonna sinensis* (25) *Prunus persica* (26) *Manglietia fordiana* (27) *Acer buergerianum* (28) *Mahonia bealei* (29) *Magnolia grandiflora* (30) *Populus Canadensis* (31) *Liriodendron chinense* (32) *Citrus reticulate*.

**Table 1 Features extracted.**

| Type | Variable | Description |
|---|---|---|
| Geometrical | $A$ | Area |
| | $P$ | Perimeter |
| | $D$ | Diameter |
| | $C$ | Compatibility |
| | $Co$ | Compactness |
| | $S$ | Solidity |
| Texture | $u$ | Median |
| | $\delta^2$ | |
| | $E_{r,\theta}$ | |
| | $H_{r,\theta}$ | Variance |
| | $HG_{r,\theta}$ | Uniformity |
| | $I_{r,\theta}$ | Entropy co-occurrence |
| | $\varphi_1$ | Homogeneity |
| | $I_1, I_2$ | Inertia |
| Morphological | | Hu1 |
| | | Ami1-Ami2 |

**Figure 3 Samples of butterflies.** (1) *Agraulis vanillae* (2) *Anthocharis midea* (3) *Ascia monuste* (4) *Danaus gilippus* (5) *Danaus plexippus* (6) *Dryas iulia* (7) *Enodia portlandia* (8) *Glutophrissa Drusilla* (9) *Heliconius charithonia* (10) *Pieres rapae* (11) *Pontia protodice.*

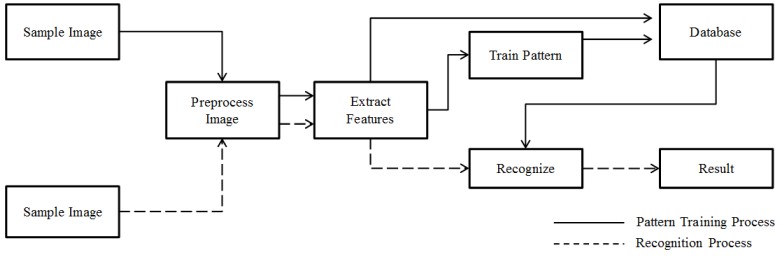

**Figure 4** System architecture.

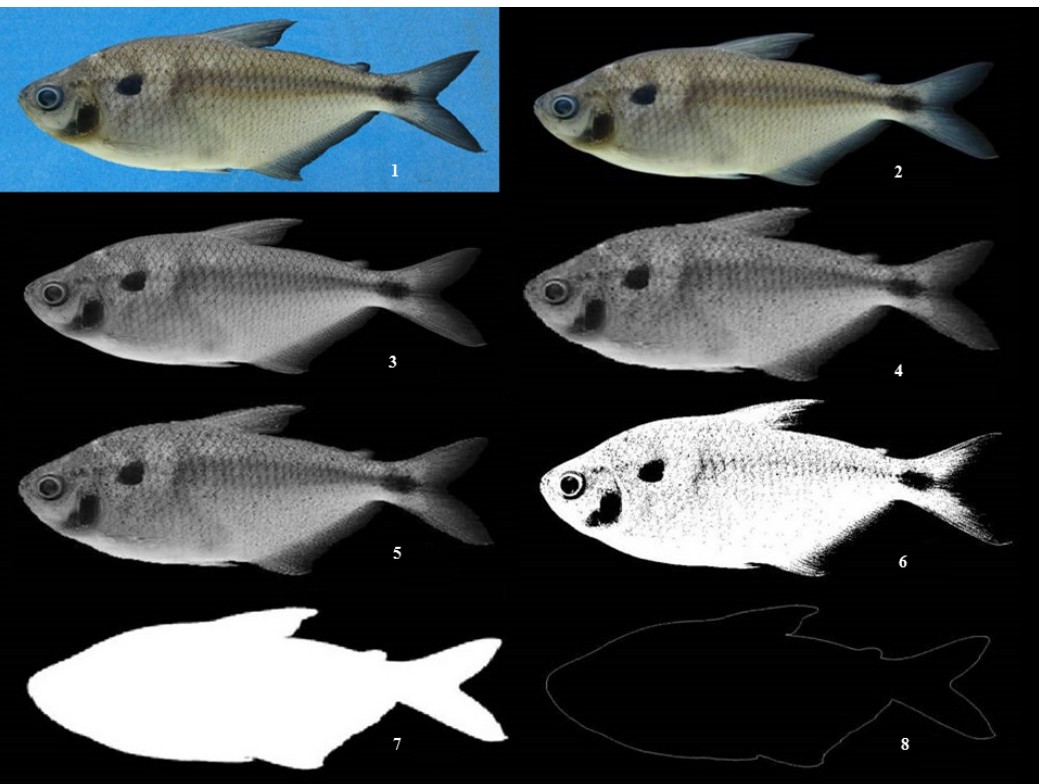

**Figure 5 Image processing.** (1) jpg image, (2) Image background was removed, (3) grayscale image, (4) smoothing filter, (5) median filter, (6) binarized image, (7) contour image (8) skeletonized image.

1-*Area* was the total number of pixels of the specimen area, and was defined as:

$$A(s) = \int_x \int_y I(x, y) dy dx$$

$I(x, y)$ depended on the limits of the shape (Fig. 5.7).

2-*Perimeter* was the number of pixels that belonged to the edge of the region (Fig. 5.8). In other words, it was the curve that enclosed a region $S$, defined as

$$P(s) = \int_t \sqrt{x^2(t) + y^2(t)} dt.$$

3-*Diameter* was the value that represented the diameter of a circle with the same area as the region. 4-*Compatibility* was the efficiency of the contour or perimeter $P(s)$ that enclosed the area $A(s)$

$$C(s) = \frac{4\pi A(s)}{P^2(s)}.$$

5-*Compactness* was the efficiency with which area $A(s)$ enclosed a speciment and was determined by $P(s)$

$$Co(s) = \frac{P^2(s)}{4\pi A(s)}.$$

6-*Solidity* was the scalar specifying the proportion of the pixels in the convex hull that were also in the region. This property was supported only for 2-D input label matrices.

## Texture

Textures are important visual patterns for homogeneous description of regions. Intuitive measures provide properties such as smoothing, roughness, and regularity (*Glasbey, 1996*). Textures depend on the resolution of the image and can follow two approaches: statistical and frequency. We used the statistical approximation in which statistical values are analyzed first order (on the histogram) and second order (on the co-occurrence matrix).

*Statistical first order* was obtained from the gray level histogram of the image. Each value was divided by the total number of pixels (area) and had a new histogram representing the probability that a determined gray level was displayed in the region of interest.

The properties obtained were:

7-*Median*

$$\mu = \sum_{x=1}^{n} xh(x).$$

8-*Variance*

$$\delta^2 = \sum_{x=1}^{n} (x - \mu^2)h(x).$$

The second order of statistics were the matrix of spatial dependence of gray levels or co-occurrence matrices. Given a vector of polar coordinates, $\delta = (r, \theta)$ we calculated the conditional probability that two properties appeared separated by a given distance $\delta, P_\delta$ using an angle $\theta$ of $-45$ and a distance $r$ equal to one pixel. The features that were extracted from this matrix were:

9-*Uniformity*

$$\sum_{x=1}^{n}\sum_{y=1}^{n} P_\delta(x,y)^2.$$

10-*Entropy co-occurrence*

$$-\sum_{x=1}^{n}\sum_{y=1}^{n} P_\delta(x,y) \log P_\delta(x,y).$$

11-*Homogeneity*

$$\frac{\sum_{x=1}^{n}\sum_{y=1}^{n}P_{\delta}(x,y)}{1+|x-y|}.$$

12-*Inertia*

$$\sum_{x=1}^{n}\sum_{y=1}^{n}P_{\delta}(x,y)(x-y)^{2}.$$

## Morphological

The morphological features were those that concentrate on the organization of pixels. They performed a comprehensive description of the region of interest. They fell into two categories: two-dimensional Cartesian moments and normalized central moments.

*The two-dimensional Cartesian moments* were variable at minor order, and were initiated at zero at higher orders. The moment of order $p$ and $q$ of a function $I(x,y)$ was defined as:

$$m_{pq}=\int_{-\infty}^{\infty}\int_{-\infty}^{\infty}x^{p}y^{q}I(x,y)dxdy.$$

$m_{pq}$ statistical moments, the parameters $p$ and $q$ denoted the order of the moment. When $p=0$ and $q=0$, which determined the center of mass or gravity of the overall function in binary images, the center of mass or gravity of the region under study was:

$$\bar{x}=\frac{m_{10}}{m_{00}}\quad\bar{y}=\frac{m_{01}}{m_{00}}.$$

The center of mass or gravity can defined the central moments that were invariant to displacement or translation of the image's region of interest defined as:

$$u_{pq}=\sum_{x}\sum_{y}(x-\bar{x})^{p}(y-\bar{y})^{q}I(x,y)\Delta A.$$

Where $\Delta A$ was the area of a pixel.

*The normalized central moments* were invariant to scale which was defined as:

$$n_{pq}=\frac{u_{pq}}{u_{00}^{\gamma}}$$

where $\gamma=\frac{p+q}{2}\ \forall p+q\geqslant 2$.

The above equations were defined by seven moments that were invariant to rotation, translation, and scale changes, known as the Hu invariant set of moments (*Hu, 1962*). In this study, we used the first Hu moment defined as:

13-*Hu1*

$$\varphi_{1}=m_{20}+m_{02}.$$

Normalized central moments were generated by related moment invariants "AMI" (*Flusser & Suk, 1993*), based on the theory of algebraic invariants and invariants under general affine transformation. We used two of the four invariants associated with discriminant character moments defined as:

14-*Ami1*

$$I_1 = \frac{u_{20}u_{02} - u_{11}^2}{u_{00}^4}.$$

15-*Ami2*

$$I_2 = \frac{u_{30}^2 u_{03}^2 - 6u_{30}u_{21}u_{12} + 4u_{30}u_{12}^3 + 4u_{21}^3 u_{03} - 3u_{21}^2 u_{12}^2}{u_{00}^{10}}.$$

These moments enable a high degree of insensitivity to noise that is not altered by rotation, translation, or staggering.

The use of the above 15 features (Table 1) characterised the structure of the individual's body, which was important for the identification at species level. We designed and realized automatic extraction algorithms to compute the values of these features so that all variables and features were calculated automatically.

## Neural network

A neural network is defined as a parallel computer model composed of a large number of adaptive processing (neural) units which communicate via interconnections with variables. A multiple layer network has one or more layers (neurons) that enable the learning of complex tasks by progressively extracting more meaningful features from the input image patterns (*Wu, 1997*). Compared to other machine learning methods, neural networks learn slower but predict faster and have very good models for nonlinear data. The simple perceptron was assigned multiple inputs but generates a single output, similar to different linear combinations that depend on input weights and generated a linear activation function (*Rosenblatt, 1958*). Mathematically, the neural network was described with the following equation:

$$y = \varphi\left(\sum_{i=1}^{n} w_i * x_i + b\right)$$

$w_i$: weight vector, $x_i$: input vector, $b$: bias activation function.

A multilayer perceptron consisted of a set of source nodes containing one or more input layer and a set of hidden-node outputs. The input signal propagated through the network layer by layer (*Zhang, Patuwo & Hu, 1998*) (Fig. 6).

The neural network structure was composed of $N$ inputs $N = [N_1, N_2, \ldots, N_n]$, a hidden layer $h$ and an output vector $S = [S_1, S_2, \ldots, S_m]$. Each $S_i$ was assessed by a single step that transformed the vector $S$ binary signal $[0, 1]$. A supervised training phase, or sigmoid activation, is based on the back propagation algorithm in which the weights and biases were updated in the direction of the negative gradient of the performance and then

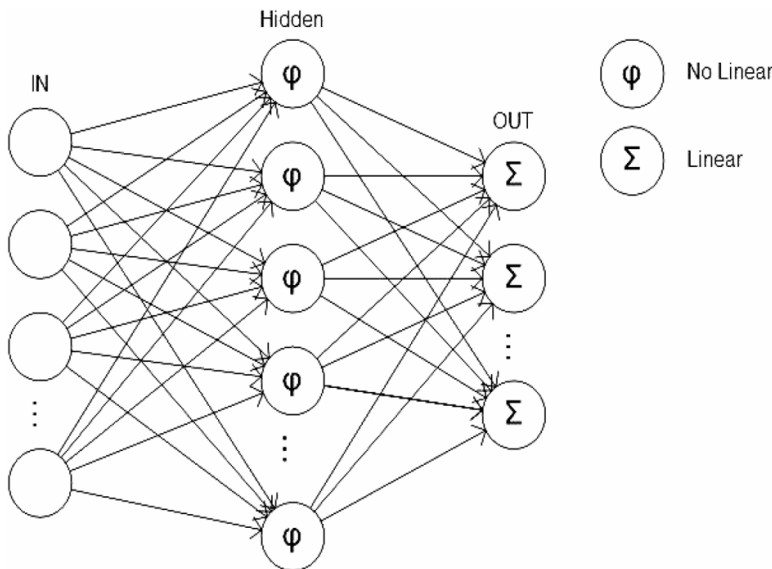

**Figure 6** Multilayer perceptron.

updated in the opposite direction (*Werbos, 1974*; *Rumelhart, Hinton & Williams, 1986*; *Parker, 1987*; *Smith & Brier, 1996*). The sigmoid activation function for the hidden layer and output layer was determined by the following equation:

$$f(x) = \frac{1}{1 + e^{-x}}.$$

In this study, the number of input neurons was determined by the number of descriptors that were available in each pattern, which in this case was 15 (see variables section). The number of neurons in the hidden layer, $h$, was experimentally determined from the error set by comparing with the general training data of the ANN. The number of output neurons was determined by the number of species classified in each database.

To determine the optimal number of neurons given a data image, the relationship between the identification success rate and the number of neurons was explored. Figure 7 shows this relationship for the different configurations considered. We finally established our networks with 200 neurons for FC-MZUSP, 180 neurons for FC-INPA, 60 neurons for FC-MNRJ, 250 neurons for FC-INVEMAR, 60 neurons for FC-CIUA, 300 neurons for FC-CRBMUV, 250 neurons for FC-MNCN, 60 neurons for FLAVIA, and 35 neurons for BUTTERFLIES (see Table 2). The number of generations (i.e., a finite set of input patterns presented sequentially) for training and testing the ANNs was variable in the different collections between 50,000 and 140,000. It is evident that a high number of neurons and generations is required to process the information of the images in each collection.

The data set was randomly divided into 60–70–80–90% training images, resulting in 40–30–20–10% test images (Table 3). The results with the highest average accuracy for species identification were networks using 80–90% training and 20–10% test images. For these tests, the declared success rate was related to the number of species identified

**Peer**J

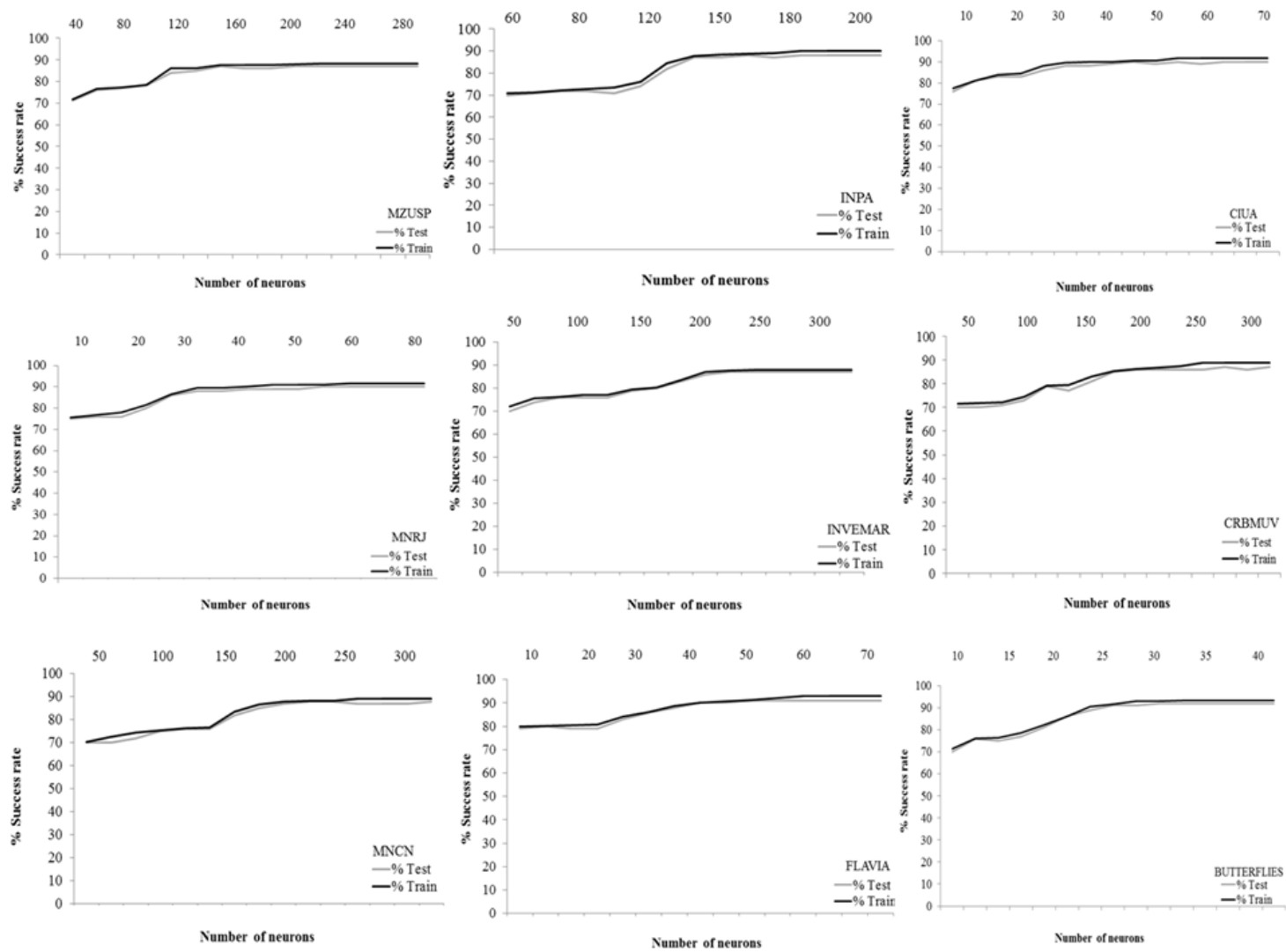

**Figure 7 Rate and the number of neurons.** Relationship between the success rate and the number of neurons for each neural network.

correctly. Recognition became more difficult with increased species number, as observed in the results from collections from MZUSP, INPA, INVEMAR, CRBMUV, and MNCN which averaged below 90% recognition.

## RESULTS

Experiments were divided into two groups: (1) images from the training group were used for building the classifications of the model; (2) images from the test group were used for the reorganization and testing of the developed model.

## DISCUSSION

Similar to previous findings (*Strachan, Nesvadba & Allen, 1990*; *Storbeck & Daan, 2001*; *White, Svellingen & Strachan, 2006*; *Zion et al., 2007*; *Novotny & Suk, 2013*), the neural network used classified species from image data. However, most other studies only

**Table 2  Parameters used in ANN.** Parameters used in neural network systems.

| Data set | Learning rate | Number of generations | Number of Hidden layers | Number of input layers | Number of output layers (# species) |
|---|---|---|---|---|---|
| FC-MZUSP | 0.2 | 95,000 | 200 | 15 | 100 |
| FC-INPA | 0.15 | 100,000 | 180 | 15 | 91 |
| FC-MNRJ | 0.25 | 78,000 | 60 | 15 | 14 |
| FC-INVEMAR | 0.3 | 84,000 | 250 | 15 | 189 |
| FC-CIUA | 0.12 | 90,000 | 60 | 15 | 33 |
| FC-CRBMUV | 0.35 | 140,000 | 300 | 15 | 172 |
| FC-MNCN | 0.2 | 110,000 | 250 | 15 | 98 |
| FLAVIA | 0.1 | 50,000 | 60 | 15 | 32 |
| BUTTERFLIES | 0.5 | 50,000 | 35 | 15 | 11 |

**Notes.**
FC, Fish collection.

**Table 3  Results of ANN.** Results of ANN tests with species tests for 15 features.

| Data set | Species | Images | Average percentage of images (Training/test) | | | |
|---|---|---|---|---|---|---|
| | | | 60/40 | 70/30 | 80/20 | 90/10 |
| FC-MZUSP | 100 | 1,718 | 76.67 | 81.34 | 83.34 | 88.31 |
| FC-INPA | 91 | 1,640 | 76.29 | 78.94 | 84.44 | 89.93 |
| FC-MNRJ | 14 | 422 | 82.62 | 87.18 | 90.56 | 91.65 |
| FC-INVEMAR | 189 | 1,703 | 76.72 | 84.03 | 86.45 | 88.08 |
| FC-CIUA | 33 | 472 | 83.08 | 86.99 | 90.19 | 91.77 |
| FC-CRBMUV | 172 | 2,392 | 77.36 | 85.21 | 87.29 | 88.85 |
| FC-MNCN | 98 | 959 | 72.34 | 86.21 | 88.15 | 89.11 |
| FLAVIA | 32 | 1,800 | 68.79 | 88.48 | 91.61 | 92.87 |
| BUTTERFLIES | 11 | 92 | 73.62 | 80.43 | 88.83 | 93.25 |

**Notes.**
FC, Fish collection.

employed databases with low levels of species richness usually spanning many different orders and families and were easily classified due to large differences in morphological characteristics. Our neural network built on the work of these networks, and required low operator expertise, costs, and response time. It also offered high reproducibility, species identification accuracy, and usability. The ANN algorithm was optimized for testing datasets with high levels of species richness; in this case 740 species (11,198 individuals) of fishes, plants and butterflies.

The predictive ability of the ANNs was affected by the high phenotypic similarity between species in our analysis. For example, small fish species such as those from the family Characidos were similar and difficult to distinguish (Annex S1, Fig. 8). The magnitude of this error came from low phenotypic differences of some species that varied only in minor details, of teeth or fin radii, which hinders classification. However, the error obtained on the neural network model was low in other taxonomic families (Table 3). Overall performance of the system achieved high accuracy and precision, with 91.65%

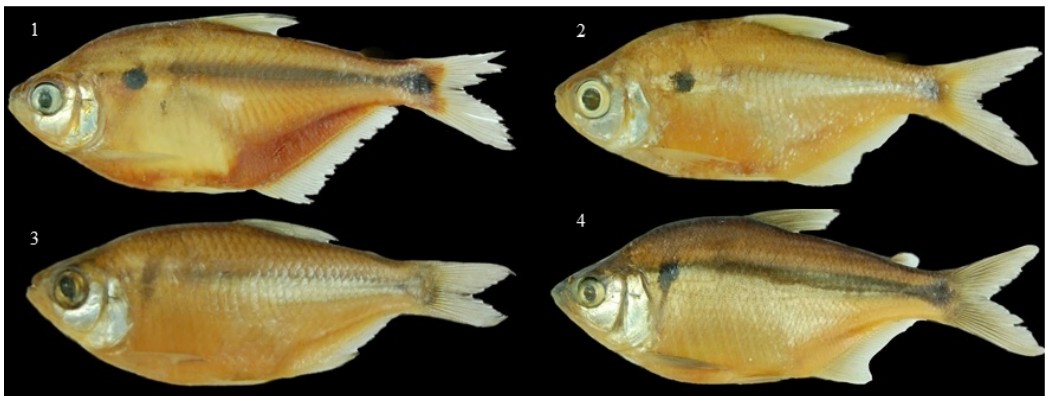

**Figure 8 An example of species confusion in the genus Astyanax.** (1) *Astyanax magdalenae*, (2) *Astyanax caucanus*, (3) *Astyanax fasciatus*, and (4) *Astyanax microlepis*.

true positive fish identifications, 92.87% plant identifications, and 93.25% butterfly identifications. The discrimination of species with a lower species number had higher success rates, possibly explained by species with very distinct morphological characteristics rather than actual number of species.

Direct observation of an individual through a taxonomic key is the most widely used technique for species recognition and classification. This technique not only assumes prior knowledge in the area of taxonomy by those who apply it, but also training and experience to achieve acceptable classification results. Training and experience are absolutely necessary for the classification specialist, who must acquire an ability to distinguish specific characteristics of the species. Therefore, we compared features of individual images with classifications of a traditional taxonomist. According to taxonomists and classification keys, characters used to discriminate species are morphological structures, color patterns, and sizes. These observations are taxonomical characteristics of individuals that depend on the particular appreciation of the taxonomist. Thus, some taxonomists may bias the value of any given characteristic, and may also require relatively more time than others to carry out the classification. Therefore, human subjectivity and time constraints may be eliminated through the use of machine based classification.

## CONCLUSIONS

The method we propose for feature extraction does not depend on variations in how a person observes individual specimens of each species, and therefore eliminates human subjectivity. For this reason, the method can be a rapid and effective species identification tool. However, a human taxonomist is still required to train the neural network defining species, and subjectivity or uncertainty is possible in this step.

The strength of this research is in its applicability to combat the "taxonomic crisis". In the past three decades, many promising techniques for fish identification have emerged. Many of them are based on genetics, interactive computer software, image recognition, hydro-acoustics, and morphometric (*Fischer, 2013*). In our study, neural networks were tested as a possible method for species identification. However, taking advantage of the fast

performance of the ANNs and the speed of modern PCs, further research should explore the applications of the ANN methodology to automate biomass estimation and real-time species classifications. This could produce useful tools for both scientific and commercial use. *Fischer (2013)* concluded that the image recognition methods were useful but their transferability and resolution are poor because species differ between geographic regions. This is a clear obstacle to future ANN development and network identification success. Our advances in this field in relation to species identification should be developed for specific geographic regions and translated into user-friendly applications.

## ACKNOWLEDGEMENTS

We thank Dr Paulo A. Buckup (Museu Nacional Rio de Janeiro), Dr José Luís Olivan Birindelli (Museu de Zoologia da Universidade de São Paulo, Sao Paulo-Brasil), Dr Rosseval Leite (Instituto Nacional de Pesquisas da Amazônia, Manaus-Brasil), Dra Gema Solís (Museo Nacional de Ciencias Naturales, Madrid-España), Dr Efrain Rubio Rincon (Colección de Referencia Biología Marina, Universidad del Valle, Cali-Colombia), and Dra Andrea Polanco (Instituto de Investigaciones Marinas y Costeras, Santa Marta- Colombia) for permission to photograph the fish, Cesar Uribe for his help on data analysis, and Jonathan Bustamante Alvarez for the photos on Fig. 1 (Universidad de Antioquia) and thanks to Benjamin Branoff, Aaron Hogan, and Paul Furumo for reviewing the English.

### Funding

Funds for this research were provided by the Universidad de Antioquia (P2010-0010 and P2011-0015). The funders had no role in study design, data collection and analysis, decision to publish, or preparation of the manuscript.

### Grant Disclosures

The following grant information was disclosed by the authors:
Universidad de Antioquia: P2010-0010, P2011-0015.

### Competing Interests

The authors declare there are no competing interests.

### Author Contributions

- Andrés Hernández-Serna conceived and designed the experiments, performed the experiments, analyzed the data, wrote the paper, prepared figures and/or tables, reviewed drafts of the paper.
- Luz Fernanda Jiménez-Segura contributed reagents/materials/analysis tools, wrote the paper, reviewed drafts of the paper.

### Supplemental Information

Supplemental information for this article can be found online at http://dx.doi.org/10.7717/peerj.563#supplemental-information.

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
