# Peer review of "Automatic identification of species with neural networks"

_PeerJ, doi:10.7717/peerj.563_

## Round 0.1 · original submission · Minor Revisions

Please attend to the referees suggestions as completely as possible unless you feel strongly that the changes are incorrect.

·

Basic reporting

1. Multiple references are not chronologically ordered, and as per reference format.
2. In the abstract line No.13 and 14 it is given as "We test the new system using our own data set that includes 740 speices .... which is contradictory to the image data used in the materials and methods

3. line 65 and 66 - sentence may be revised.
4. Feature values are not given. This may be given in a table format
5. In the Figures, for listing, numbering may be adopted instead of alphabets (because it is lengthy)

6. Figure legends - not clear
7. in line 99 Textures depend on
8. Line No. 111, 112, 128, 130 - symbol is not in line - need alignment
9. In line No 175 Brackets may be removed
10. Web References - not given as per the format

Experimental design

No comments

Validity of the findings

1. Feature values are not given. This should be given in a table format as main data or supplementary data
2. Model website address should be given

Comments for the author

Required revision.

Reviewer 2 ·

Basic reporting

The manuscript still contains some problems in English writing. Thus they cause cofusions regarding to exact meanings in some parts of it.
So one round of edition is strongly needed for improvement.
The standard vocaburary for the repetitive "study region" or "region of study" might be "region of interest" which is being used as standard words.
Followings are some examples of what I talk about:
68: Image processING
118: organization OF pixels
126~127 this sentense may be rephrased as like, "With Center of Mass/Center of Gravity, central moments, which are invariant to displacement or translation of the Region of Interest, can be defined as:"
129: as defined as -> which is defined as
141: These moments enable a high degree of insensitivity to noise ...
153: extracting progressively -> progressively extracting

the other miscellaneous mistakes are skipped hereafter.

Experimental design

There is no comparison with other methods with respect to the accurarcy of the current methods. Although the authors referred to many relevant papers including the Gaston 2001, but this paper did not provide any comparative results with other methods. I think the authors, at least, be able to comment on how their methods are fast, do not need manual work, or any other advantageous points compared to other previous methods. So I recommned such comparisons. The authors do not need to be worry about any inferior thing in their result. Just frank comparision will be helpful for readers.
Followings are ambigous parts. The authors can improve them with more explict explanations.
line 88: Area feature, Diameter, Perimeter... there are still ambiguities.
Fig. 5 shows image processings. I thinkg each process deals with different feature extraction. For example, Fig. 5g is for the extraction of area, Fig. 5h for perimenter and so on. So additional explanations are needed for readers.

There are different machine learning techniques like SVM, ANN, Dicision tree. I don't see why the authors used ANN among them. Please specify why you chose the ANN instead of others. Readers probably want to hear the reason.

around line 202: there is an explanation about some errors in original identification of species. I wonder how much proportion this kind of errors is. Perhaps such errors as photos will be a appealing example for your explanation, if you provide them. Otherwise it will be wise if you remove such errors and then reanalyze your input data, althugh you showed higher accuracy in the cases of plants and butteflies that did not include such errors.

line 207-210: It is somewhat strange that the smaller the individuals the higher accuracy. This is anti-intuitive for the principle of ANN classification. Perhaps there is a reason for it. For example the species with small number of data might be more distinct than the others. I am not sure for it. So please address this problem carefully.

Validity of the findings

No Comments.

Comments for the author

The methods of automatic identification of various speics including different plants and animals are quite well explained as such that other researchers could reproduce the same result as the present paper. And there seems to be no problem such as researchers may need to specify particular areas to improve better feature extraction with naked eyes or to iterate adjustments until good performance is reached. Therefore generally this paper is acceptable in both aspects of the technical point and high success rate of automatic classification of relatively huge image data including many species.

---

## Round 0.2 · Minor Revisions

Please provide more specific details in your response to the referees comments. Just saying "RESPONSE: We agree with the reviewer and thus we include in manuscript" is not helpful, as well as being bad English. Please also have the MS reviewed for the quality of its English.

---

## Round 0.3 · Minor Revisions

This is very nice work. Well done. There are two aspects I'd welcome your response to. The first regarding the data availability, and the second to revise the Results text.

Both referees and I feel the underlying data should be made public. It takes some reading to understand what 'features' are but they are central to the study. Would their publication help future studies or are they so peculiar to this one that they would be of no use to others?

To a taxonomist the 'features' would be metrics that define characters that distinguish species. I think it would be helpful to provide a table of the kinds of 'characters' discovered by the analysis because they may indicate better ways to idnetify species than used by taxonomists at present. That is, how can this method better inform guides to species identificaiton?

The Results should be separated from the Discussion. The first few paragraphs are not needed as they just tell the reader what it is going to be told. Statements like "Fig. 7 shows this relationship for the different configurations considered." are inconcise. It does not tells the reader what you found. Instead, state the key finding and cite the figure or table. In that way the reader know swhat you found and what was done and where to see the data.

---

## Round 0.4 · accepted · Accept

Thank you for the revised text and adding the dataset. The Discussion especially could merit better English. It would help if any reference to the work done (whether reported in Methods, Results or Discussion, was in the past tense; as it clearly was in the past.

I highly recommend the text is reviewed by somebody good in English to improve its clarity. e.g., This sentence does not make sense "Therefore, by comparing extracted features of individual images with classifications of a traditional taxonomist, would be that the algorithm proposed uses color as variations present in each of the individuals that are acquired by the normalized central moments of entropy and inertia." Reducing technical jargon would help clarity.